# Determination of Total Sennosides and Sennosides A, B, and A_1_ in Senna Leaflets, Pods, and Tablets by Two-Dimensional qNMR

**DOI:** 10.3390/molecules27217349

**Published:** 2022-10-29

**Authors:** Serhat Sezai Çiçek, Calisto Moreno Cardenas, Ulrich Girreser

**Affiliations:** 1Pharmazeutisches Institut, Abteilung Pharmazeutische Biologie, Christian-Albrechts-Universität zu Kiel, Gutenbergstraße 76, 24118 Kiel, Germany; 2Pharmazeutisches Institut, Abteilung Pharmazeutische und Medizinische Chemie, Christian-Albrechts-Universität zu Kiel, Gutenbergstraße 76, 24118 Kiel, Germany

**Keywords:** *Senna alexandrina*, *Cassia angustifolia*, *Cassia acutifolia*, quantitative NMR, surrogate standard, aloin, quality control, laxatives, anthrone glycosides, anthranoids

## Abstract

In the present work, a two-dimensional qNMR method for the determination of sennosides was established. Using band-selective HSQC and the cross correlations of the characteristic 10–10’ bonds, we quantified the total amount of the value-determining dianthranoids in five minutes, thus, rendering the method not only fast, but also specific and stability indicating. The validation of the method revealed excellent accuracy (recovery rates of 98.5 to 103%), precision (RSD values of 3.1%), and repeatability (2.2%) and demonstrated the potential of 2D qNMR in the quality control of medicinal plants. In a second method, the use of 2D qNMR for the single analysis of sennosides A, B, and A_1_ was evaluated with acceptable measurement times (31 min), accuracy (93.8%), and repeatability (5.4% and 5.6%) for the two major purgatives sennoside A and B. However, the precision for sennoside B and A_1_ was not satisfactory, mainly due to the low resolution of the HSQC signals of the two compounds.

## 1. Introduction

Leaves and fruits of *Senna alexandrina* Mill. (Fabaceae, Syn. *Cassia acutifolia* and *Cassia angustifolia*) have a long history of medicinal use, with first mentions by Arabic writers in the 9th century [1,2]. After having been recognized as an effective and tolerable laxative in the medieval period, powders and preparations of senna have been widely used as purgatives and are nowadays classified as well-established herbal medicinal products by the European Medicines Agency [2,3,4]. Sennosides A and B (and A_1_) are the main purgative agents, which are reduced to rhein anthrone in the human intestine [5]. Sennosides are considered among the most important pharmaceutical products of plant origin [6], a fact that is also reflected by methods and patents reporting their preparation and isolation from natural sources [7,8]. Sennosides A, B, and A_1_ are homodimeric hydroxyanthracene diglycosides, linking two rhein 8-glucoside molecules in position 10, while sennosides C, D, and D_1_ are heterodimers, consisting of one molecule of rhein 8-glucoside and one molecule of aloin 8-glucoside (Figure 1).

In the European Pharmacopeia, three monographs on senna can be found, which are either the leaflets, pods, or a standardized dry extract containing 5.5 to 8.0% hydroxyanthracene glycosides (calculated as sennoside B) [9]. Until the 9th version of the Pharmacopeia, two types of senna pods were listed, which were Alexandrian pods and Tinnevelly pods requiring minimum amounts of 3.4% and 2.2% sennosides, respectively [10]. For the leaflets instead, only one type, with a minimum amount of 2.5% sennosides, was monographed. The quality control of all drugs was accomplished with a photometric assay determining the total amount of hydroxyanthracene glycosides after aqueous extraction of the drug and removal of the aglycones by liquid–liquid partitioning with chloroform. The remaining glycosides were subsequently cleaved into their monomers and hydrolysed into their aglycone forms, which were measured at 515 nm. Another photometric assay was reported by Brendel et al. [11], which was also achieved by aqueous extraction and separation of the aglycones. However, in this assay, the total sennosides were measured without oxidative cleavage into their monomeric forms, only hydrolyzing the sugar moieties.

Starting from the 10th version, the European Pharmacopeia introduced an HPLC-UV assay for the quality control of senna pods and leaflets (but not the dry extract), summarizing the amount of eight hydroxyanthracene glycosides to a minimum amount of 2.0% (for both drugs) [9]. Further HPLC-UV assays for the quantitation of sennosides were reported by Metzger et Reif [3] and Bala et al. [12], while LC-MS and NMR were successfully applied for the analysis of the chemical composition of *S. alexandrina* and other *Senna* species [1,6,13]. Especially for stability evaluation of senna drugs and products, chromatographic methods are more useful than photometric techniques because the transformation products show absorption at the same wavelength as sennosides [14]. This is even more important as sennosides are prone to both reductive and oxidative cleavage of the 10–10’ bond [15]. Apart from their low stability, the availability of sennosides as reference standards is an issue because they have to be purchased at high costs. Still, they are essential for the use of chromatographic methods.

In this work, we applied two-dimensional quantitative NMR (2D qNMR) for the determination of the total sennoside content. This was achieved by integration of the H-10 and H-10’ proton cross signals and thus by using the critical positions for degradation processes. Therefore, our method is equally suited for stability evaluations. Two-dimensional qNMR techniques have been successfully employed for the analysis of complex mixtures [16,17,18,19,20], but their use is often avoided due to long measurement times. Another reason is the need of reference standards, because 2D qNMR (in contrast to 1D qNMR) is an indirect quantitation method [21,22,23]. For both of the mentioned shortcomings, solutions were presented previously [24,25] and have also been applied in the present study. Using band-selective heteronuclear single quantum correlation spectroscopy (bs-HSQC), the total amount of sennosides was determined in approximately five minutes, far below the measurement times of photometric or conventional chromatographic assays. By applying aloin as a surrogate standard, we furthermore avoided the need of original reference material (Figure 2).

In the same manner, a second quantitative 2D qNMR method for the determination of the major sennosides A, B, and A_1_ was established, and both assays were subsequently evaluated for the analysis of senna leaflets, pods, and commercial preparations.

## 2. Results and Discussion

### 2.1. Method Development

#### 2.1.1. Background

The major aim of this study was to establish an accurate, fast, and meaningful quantitation method for the value-determining dianthrone glycosides present in the leaflets and pods of *S. alexandrina*. qNMR in general and especially two-dimensional methods are ideal tools for the quantification of specific structural features and thus allow for the determination of the total amounts of particular constituents. Both 1D and 2D qNMR methods have been successfully applied for the quantitation of the total amounts of pharmaceutically or nutritionally relevant constituents by targeting characteristic structural features, i.e., the H-19 *exo*-proton of cycloartanoids [26] or the 11α-hydroxy group of mogrosides, with the latter feature at the same time quantifying the biological activity [27]. In addition, in the case of the value-determining dianthrones in senna pods and leaflets, such a particular feature exists, namely the 10–10’ bond linking the two monomers. This structural feature thus allows for the discrimination of sennosides from other anthrone glycosides present in the drugs and preparations. Moreover, this feature is ideally suited for stability evaluations, as sennosides are prone to both oxidative and reductive cleavage [15].

In DMSO-*d*_6_, which is one of the most suitable qNMR solvents for plant extracts, positions 10 and 10’ show proton shift values of around 5 ppm in DMSO and are thus in the range of anomeric sugar protons [8]. Because of the possible overlap of the H-10 and H-10’ signals with the anomeric sugar signals of sennosides or other molecules contained in the extract, development of a 1D qHNMR method was not taken into account. In contrast to the proton shifts, the ^13^C shift values for the respective positions are found at about 54 ppm, and hence low frequency shifted compared to anomeric sugar signals. This is most evident when looking at the HSQC diagrams of sennosides A, B, and A_1_, in which the C–H cross correlations are far off the diagonal line (Appendix A). The co-occurrence of rather high frequency shifted proton signals with relatively low carbon shift values renders the HSQC experiment the ideal technique for the quantitation of the total amount of sennosides. This can be achieved by the integration of a single cross peak, as the respective methine groups 10 and 10’ show nearly identical chemical shifts in sennosides A, B, and A_1_. Still, with also aglycone forms of the sennosides, namely sennidines, being present in the drug, interferences with the compounds of interest had to be expected and be eliminated.

#### 2.1.2. Sample Preparation

Different approaches are possible to remove undesired aglycones. The two most popular methods are pre-extraction with a more lipophilic solvent or the use of liquid–liquid partitioning. In the two reported photometric assays on senna, aglycones were removed either with chloroform or diethyl ether, both after extracting the drug with water [10,11]. Stoll et al. suggested the pre-extraction of the drug with a mixture of chloroform and ethanol (93:7) before extracting the sennosides. As for the present method, pressurized solvent extraction was applied, for which the drug powder was extracted in cartridges under high pressure and temperature; pre-extraction appeared more convenient. In order to avoid the toxic solvent chloroform, other solvents were studied for this purpose, such as acetone, ethyl acetate, and dichloromethane–methanol (93:7). Dichloromethane–methanol (93:7) and ethyl acetate both successfully extracted the sennidines while maintaining the sennosides in the drug material. Acetone, in contrast, additionally extracted small amounts of glycosides. Being less problematic in terms of toxicity and environment, ethyl acetate was chosen for pre-extraction. For the subsequent extraction of the sennosides, different mixtures of methanol and water were studied, also adding hydrochloric acid (0.1% *v*/*v*) or ammonia (0.1%), and varying extraction temperatures. Finally, a mixture of methanol–water (7:3) and a temperature of 80 °C were chosen for the extraction of the analytes.

In the next step, the sennosides were enriched by solid-phase extraction (SPE). Using strong anion exchange columns with reversed-phase properties, neutral and basic compounds were effectively eluted and acidic compounds, such as the sennosides, were maintained. The SPE extraction followed in principle the method of Yamasaki et al., who separated sennosides A and B by washing the column with water, methanol, and 1% acetic acid in methanol [28]. However, in order to not elute sennosides C, D, and D_1_ the last washing step was omitted and all sennosides were eluted together using methanol–water–formic acid (70:30:2). All extraction and SPE separation steps were monitored using UHPLC-MS for qualitative analysis and HPLC-UV for quantitation of the analytes. A conventional HSQC diagram of the final SPE extract in DMSO-*d*_6_ is depicted in Figure 3.

#### 2.1.3. NMR Measurements for Total Sennosides

Conventional HSQC is one of the standard measurements for the structure elucidation of organic compounds, giving signals for C–H correlations and thus information on all C–H bonds in the molecule. In addition, HSQC can be applied for quantitative purposes; however, depending on the magnetic field strength of the instrument and the analyte concentration, long measurement times are needed. One strategy for reducing the measurement time is using non-uniform sampling, which means that only a randomly chosen number (e.g., 50%) of all increments of the 2D NMR spectrum are measured [29]. Another efficient way to overcome the disadvantage of long measurement times is the use of band-selective pulse programs. Thereby, the spectral width of the ^13^C channel is drastically reduced to a necessary, but still applicable range. Thus, measurement times are minimized without causing spectral folding. A detailed comparison study on the use of bs-qHSQC was conducted on the quality control of aloe, where we could show that equal results were obtained with band-selective approaches [24]. For the determination of the total sennosides, we selected a ^13^C spectral range of 25 ppm (53.5 ± 12.5 ppm) in order to record both the analytes’ signals and the signal of the surrogate standard aloin (having a ^13^C shift value of 44.5 ppm) (Figure 4). Maintaining the same acquisition parameters, the reduction in the spectral range from 165 ppm (conventional qHSQC) to 25 ppm (bs-qHSQC) for the present method led to a six-fold reduction in the measuring time, which finally was 5 min and 4 s.

Another measure that can be applied in qHSQC is delay time adaption. Usually, routine HSQC spectra are recorded using delay times corresponding to the ^1^JCH coupling constant of 145 Hz, thus being a compromise between aliphatic and aromatic signals, usually having values between 110 and 175 Hz, respectively. Adaption of delay times for the respective analytes increases signal intensity and can be a helpful tool in the analysis of minor concentrations of analytes or if different concentrations are measured within one sample [25]. Coupling constants were determined using HSQC experiments without decoupling and resulted in 135 Hz for aloin (CH-10) and 141 Hz for the sennosides (CH-10 and CH-10’) (Appendix A). Consequently, delay times for the determination of the total sennosides were adapted for the ^1^JCH coupling constant of 138 Hz. 

#### 2.1.4. Quantification of Sennosides A, B, and A_1_

The second aim of this study was to examine if the major sennosides A, B, and A_1_ could also be determined selectively beside each other. An overlay of the HSQC diagrams of the isolated constituents showed that, in the range of 7.2 to 7.8 ppm (^1^H) and 134 to 136 ppm (^13^C), three separate signals occur, while most of the other signals show an overlay (Appendix A). This is supported by the NMR data for the three constituents and the respective CH-6 position in the molecule. Subsequently, a second bs-HSQC method for the quantification of the three sennosides was developed with the following adjustments.

For solid-phase extraction, twice the volume of extract was applied to guarantee sufficient analyte concentrations (especially for sennoside A_1_). In addition, monocarboxylic sennosides C, D, and D_1_ were separated using 1% acetic acid in methanol before eluting the dicarboxylic sennosides, as reported by Yamasaki et al. [29]. HPLC chromatograms of the senna crude extract and the solid-phase extract are shown in Appendix A. Coupling constants for the observed groups of both aloin and the three sennosides were determined with 163 Hz, and delay times were adjusted correspondingly. Because aloin and the sennosides show similar chemical shifts in position 6 (Appendix A), the ^13^C width for the band-selective measurement was set to only 10 ppm (and thus a 15-fold reduction in measuring time). However, the analysis of single components instead of the sum of the sennosides and the low concentration of sennoside A_1_ required a higher number of scans (32), resulting in a final measurement time of 31 min, which is in the range of conventional HPLC methods. Figure 5 and Figure 6 show conventional and band-selective HSQC diagrams of the solid-phase pod extract used for the quantitation of the single sennosides.

Both methods were subsequently used for the evaluation of senna leaflets, pods, and tablets and validated for linearity, precision, accuracy, and limit of quantification.

### 2.2. Method Validation

#### 2.2.1. Linearity and Quantitation Limit

For the validation of linearity, the calibration curves of two different cross correlations of aloin were established. Thereby, the correlation of CH-10 was used to obtain an external calibration curve for the total sennosides determination, while the correlation of CH-6 was employed to establish the calibration curve for the quantification of sennosides A, B, and A_1_. Concentration levels and the resulting regression equations were calculated in mmol/L of sennosides (Table 1). 

In the same manner, the limits of quantitation were obtained, which were determined with 0.500 mmol/L for the total sennoside content and 0.875 mmol/L for the concentrations of the single sennosides. Both LoQ concentrations were part of the respective calibration curves, which showed *R*^2^ values of 0.9955 and 0.9956, respectively, and thus exhibited good linearity over a wide concentration range.

#### 2.2.2. Precision and Repeatability

Both precision and repeatability were assessed for the total content (calculated as sennoside B) and also separately for the three sennosides A, B, and A_1_ using integration ranges for both determinations (Table 2). In two-dimensional qNMR, peak intensities can be used; however, especially for the analysis of the total content, integration ranges are more suitable. This becomes evident when looking at Figure 4, where the cross correlations of positions CH-10 and CH-10’ are reflected by two signals. Thereby, the higher shifted signal in the proton dimension results from sennoside A (5.01 ppm) and the lower shifted signal from sennoside A_1_ (4.96 ppm). The cross correlation of sennoside B, in contrast, is contained within both of the depicted signals (4.95 and 5.01 ppm). As no other signals were observed in the immediate surroundings and no NMR data for other sennosides are available, cross correlations from the latter compounds were expected in this range. Though the cross correlations of sennosides result in two signals, their marginal differences in both proton and carbon NMR shifts allowed for very narrow integration ranges, thus limiting the integration of unnecessary noise. This clean integration is reflected by the low relative standard deviation in the repeatability measurement, which was determined as 2.2%. With RSD values of 3.1% for both intra-day and inter-day precision, the sample preparation, and thus the whole assay, was found to be precise and the developed method to deliver reproducible results. 

Precision measurements for the content of the single sennosides yielded a much-differentiated picture between the three constituents. The assessment of repeatability resulted in relative standard deviations of 5.4% (sennoside A), 5.6% (sennoside B), and 10.6% (sennoside A_1_). These RSD values are reasonable given the fact that lower overall concentrations were measured. This accounts even more for sennoside A_1_, which shows a concentration of one-third to one-fourth of the other two sennosides. Intra-day and inter-day measurements, however, did not give this clear picture. Here, sennosides A and A_1_ in principle followed the pattern observed for the total content assay and showed somewhat higher RSD values, reflecting the sample preparation. These values were determined with 7.9% (intra-day) and 6.3% (inter-day) for sennoside A, and 13.6% and 13.9% for sennoside A_1_, respectively. For sennoside B, instead, the observed RSD values amounted to 11.6% (intra-day) and 10.0% (inter-day) and thus were unreasonably high. Even more so, sennoside B is the major sennoside contained in the drug. This phenomenon can be explained when looking at Figure 6, where the signals of the three sennosides are depicted. Same as for positions 10 and 10’, respectively, sennosides A and A_1_ showed one signal for the cross correlation of CH-6 and CH-6’. This allows for narrow integration ranges and reproducible results. Sennoside B, in contrast, shows two signals, which this time are also separated in the ^13^C dimension. Therefore, wider integration ranges would have been necessary, but were hindered by the close proximity of the signals to those of sennoside A_1_.

#### 2.2.3. Accuracy

Accuracy of the total sennoside assay was evaluated using commercially available senna leaflets and retard tablets (Table 3). The certificate of analysis given along with the senna leaflets indicated an amount of 2.35 ± 0.15% hydroxyanthracene glycosides, calculated as sennoside B. Analysis of the plant material with our own assay showed a value of 2.42 ± 0.08%. The amount of sennosides in the commercial retard tablets was specified with 13 mg per tablet. Here, our assay showed comparable results as well, giving an amount of 12.77 ± 0.85 mg per tablet. Thus, our methods showed excellent accuracy with recovery rates of 103% (drug) and 98.5% (tablets), respectively. The accuracy of the single sennoside assay was evaluated by comparing the single amounts of sennosides A, B, and A_1_ obtained in the precision measurements to the sum of the three compounds measured with the total content assay. This was accomplished by using a sample worked up for single sennoside analysis containing only dicarboxylic sennosides. The determined amount was 2.377 ± 0.086%. Taking the sum of the sennoside contents from the inter-day measurements (2.230 ± 0.197%), which are the most reliable values due to the highest number of analyses (12), a recovery rate of 93.8% was calculated. 

#### 2.2.4. Specificity and Selectivity

Certainly, one of the main advantages of two-dimensional qNMR is its specificity. Using chemical shift values resulting from characteristic functional groups, structural features can be quantified specifically. This accounts even more so for heteronuclear 2D qNMR, where also ^13^C shift dispersion is employed. Often, such structural features circumscribe compound classes and can then be used for their quantitation. In our method, a characteristic structural feature was used for the analysis of the target analytes, which is the 10–10’ bond linking two anthrone glycosides. The resulting dianthrone glycosides, namely sennosides, are the major purgative agents contained in senna drugs and products analyzed in this study. Using this structural feature for quantification is probably the most specific way for their determination since due to the characteristic chemical shift values of these positions, no interferences with other signals are expected.

So far, sennosides have been either determined by photometric or chromatographic techniques, which both have their strengths and limitations. Photometric assays are easy to accomplish with regard to necessary equipment, requiring a simple spectrophotometer and a laboratory for the chemical work-up. Their major drawback is the lack of specificity, which renders them unusable for, e.g., stability evaluations, as degradation products often show a similar UV absorption. Likewise, adulterations with similar components might lead to unwanted co-quantification. Chromatographic methods, instead, are better suited for the detection of adulterations, especially when equipped with MS detectors. The same accounts for stability evaluations, where the analytes in general can be studied beside degradation products. With our assay, the stability of drugs and drug products can be observed easily because the decomposition of sennosides results in a cleavage of the 10–10’ bond [15], and the remaining degradation products can therefore not be co-quantified.

Our second method, which we used for quantifying sennosides A, B, and A_1_, furthermore demonstrates that very similar components can be determined side by side. This is achieved by using the CH-6 cross peaks, which afford different cross signals for each of the three diastereomers. Thus, obtaining several signals for each analyte (as usual in NMR) does not only increase the complexity of the analysis, but can also add towards selectivity of the respective method. This was demonstrated previously by the successful 2D NMR quantitation of compounds that were overlooked by the LC and LCMS methods [20].

## 3. Materials and Methods

### 3.1. Materials and Reagents

Dried and cut leaflets (Lot 26094) and pods (Lot 22421) were obtained from Alfred Galke GmbH (Bad Grund, Germany). Retard tablets (Sennalax^®^, Lot 2014211) were bought in a local pharmacy. SPE cartridges (Chromabond HR-XA, 85 µm, 3 mL/500 mg) and a 70 mL reservoir column were obtained from Macherey-Nagel GmbH (Düren, Germany), and cartridges for flash chromatography (Flashpure Ecoflex Silica SL 50 μm irregular 10 g, Lot 150317B) were ordered from Büchi (Flawil, Switzerland). Dimethyl sulfoxide-*d*_6_ (99.80%, lot S1051, batch 0119E) for NMR spectroscopy was purchased from Eurisotop GmbH, Saarbrücken, Germany, and conventional 5 mm sample tubes were obtained from Rototec-Spintec GmbH, Griesheim, Germany. LCMS-grade acetonitrile, gradient-grade methanol, analytical-grade solvents, and TLC plates (silica gel 60 F254) were purchased from VWR International GmbH, Darmstadt, Germany. Hydrochloric acid (34%) and trifluoroacetic acid (TFA) were obtained from Carl Roth GmbH (Karlsruhe, Germany), acetic acid was purchased from Thermo Fisher Scientific Inc. (Waltham, MA, USA), and LCMS-grade formic acid was purchased from Merck (Darmstadt, Germany). Water was doubly distilled in-house.

### 3.2. General Experimental Procedures

Pressurized solvent extraction was performed with a Speed Extractor E961, and flash chromatography was accomplished with a PrepChrom C-700 (Büchi, Flawil, Switzerland). HPLC analyses were performed on an Ultimate 3000 instrument equipped with an HPG-3400SD pump, a WPS-3000SL autosampler, a TCC-3000SD column heater, and a VWD-3400RS variable wavelength detector (Thermo Fisher Scientific Inc., Waltham, MA, USA). A Synergi Max RP column (150 × 4.6 mm, 4 µm particle size) was used for separation (Phenomenex, Aschaffenburg, Germany). The solvent system, gradient, and flow rate were as follows: solvent A: 0.025% (*v*/*v*) TFA in water, solvent B: acetonitrile, flow rate: 1.0 mL/min, gradient: 5% B in 20 min to 25% B, in 10 min to 95% B, in 10 min to 95% B, followed by 10 min with 5% B (postrun). The column temperature was 40 °C. LCMS measurements were conducted on a Nexera X2 instrument consisting of an LC-30AD binary pump, an SIL-30AC autosampler, a CTO-20AC column oven, and an SPD-M30A diode-array detector (Shimadzu, Kyoto, Japan). Photometric measurements were conducted on a Shimadzu UV Mini 1240 spectrophotometer.

NMR spectra were recorded using a Bruker Avance III 400 NMR spectrometer (Bruker, Rheinstetten, Germany) operating at 400.33 MHz for the proton channel and at 100.66 MHz for the ^13^C channel by means of a 5 mm PABBO broad-band probe with a z gradient unit. Measurements were performed at 298 K; the temperature of the probe head was calibrated with a methanol-*d*_4_ solution and was regulated within 0.1 K during the measurements. For each sample, automatic tuning and matching of the probe was performed, as was automatic shimming of the on-axis shims (*z* to *z*^5^). Bruker Topspin software 3.6.0 was employed for spectra recording. The excitation pulse was determined with 9.0 µs for this solvent and sample type; this value was used throughout. For the determination of coupling constants, conventional full-scan HSQC spectra without decoupling were recorded.

### 3.3. Isolation of Reference Standards

The isolation of sennosides A, B, and A_1_ initially followed the procedure described by Stoll et al. [30], with some modifications. Briefly, 100 g of dried and powdered senna pods were extracted four times for 10 min, each using 375 mL of a mixture of dichloromethane and ethanol (93:7). The remaining plant material was subsequently extracted with methanol using the same extraction procedure. The methanolic solution was then reduced in a rotary evaporator until precipitation of a yellow solid occurred (approximately 200 mL). The solid was collected, and 30 mL of 10% calcium chloride in methanol were added to the solution, forming another precipitate. After the repeated evaporation of the solvent and the subsequent treatment with 3.5 mL of 10% triethylamine in methanol, a third and fourth precipitate, respectively, were obtained. After TLC analysis of the precipitates, two main fractions with 340 (I) and 350 (II) mg were further processed. Fraction I was subjected to silica gel flash chromatography using a mixture of *n*-propanol-ethyl acetate–water and a gradient of 45:45:10 to 35:35:30 in 60 min, to 40:20:40 in 30 min, and to 40:20:40 in another 30 min. the detection wavelength was 270 nm. Fraction II was separated with the same solvent mixture using a gradient of 50:50:0 to 35:35:30 in 60 min and to 20:40:40 in 30 min. Subfractions were combined according to the TLC results, yielding 77 mg of sennoside A, 69 mg of sennoside B, and 13 mg of sennoside A_1_. The isolation of aloin was performed as described by Girreser et al. [24].

### 3.4. Determination of Total Sennoside Content

A total of 1.00 g of dried and powdered plant material was extracted by pressurized solvent extraction with 5 mL of ethyl acetate at 80 °C and 100 bar. Ethyl acetate was removed, and the material was extracted six times with 5 mL of methanol 70%. The solution was diluted to 50 mL with methanol before 20 mL were subjected to anion exchange SPE using the following procedure: (i) column conditioning with 20 mL of methanol followed by 20 mL of water, (ii) sample application, (iii) washing with 20 mL of water followed by 20 mL of methanol, and (iv) elution of sennosides with 30 mL of methanol–water–formic acid (70:30:2). The eluate was evaporated to dryness, and the residue was dissolved in 600 µL of DMSO-*d*_6_.

For the quantification of the total sennosides, band-selective heteronuclear single quantum correlation spectroscopy (bs-HSQC) measurements were performed using the phase-sensitive shsqsctgpsi2.2 pulse program of the manufacturer’s pulse program library with a band-selective shaped ^13^C refocusing pulse and ^13^C GARP decoupling. The ^1^H spectral range was (5.20 ± 6.50) ppm, a spectral range of 5200 Hz, whereas the ^13^C spectral range was (53.5 ± 12.5) ppm, corresponding to 2500 Hz. The shape form Q3.1000 was chosen for the selective refocusing pulse, and in order to achieve selective excitation over the frequency range of 2500 Hz (25 ppm), the length of the pulse was determined to be 1379.2 μs with a power of 0.6166 W (2.1 dB) for the probe head in use. For the ^1^H channel, 1024 data points were collected, whereas for the ^13^C channel, 52 data points were set. Using non-uniform sampling, only 75% of the increments were recorded. The acquisition time was 0.11 s, and the inter-scan delay was set to 3.0 s. The relevant delays in the pulse program were set corresponding to a coupling constant of ^1^*J*CH = 138 Hz, that is d_4_ = 1.8116 ms and d_24_ = 0.9058 ms. With 16 dummy scans and 2 scans per increment, the total acquisition time for this HSQC experiment was about 5 min.

### 3.5. Determination of Sennosides A, B, and A_1_

For the determination of the single sennosides, the same extraction procedure was used with two modifications of the solid-phase extraction: (i) Instead of 20 mL, 40 mL of sample was applied. (ii) After the washing steps, monocarboxylic sennosides (e.g., C, D, and D_1_) were eluted with 1% acetic acid in methanol before eluting the dicarboxylic sennosides A, B, and A_1_ with the mixture mentioned above. 

For quantification, the same pulse program was used; only relevant changes in the parametrization are given: the ^1^H spectral range was (5.20 ± 6.50) ppm, a spectral range of 5200 Hz, whereas the ^13^C spectral range was (136.0 ± 5.0) ppm, corresponding to 1000 Hz. In order to achieve selective excitation over the frequency range of 1000 Hz (10 ppm), the length of the pulse was determined to be 3448 μs with a power of 0.09868 W (10.1 dB) for this probe head. For the ^1^H channel, again, 1024 data points were collected, whereas for the ^13^C channel, 36 data points were set. Using non-uniform sampling, only 50% of the increments were recorded. The acquisition time was 0.11 s, and the inter-scan delay was set to 3.0 s. The relevant delays in the pulse program were set corresponding to a coupling constant of ^1^*J*CH = 163 Hz, that is d_4_ = 1.5337 ms and d_24_ = 0.1669 ms. With 16 dummy scans and 32 scans per increment, the total acquisition time for this HSQC experiment was about 31 min.

### 3.6. Spectroscopic Analysis

To quantify the sennosides by band-selective HSQC, two different cross peaks of the surrogate standard aloin were selected, and 7-point calibration curves ranging from 0.425 to 19.325 mg/mL were established (Figure 2, Table 1). All solutions were prepared with DMSO-*d*_6_. Data processing was performed using the Topspin software. The raw data was processed for both methods as follows: zero filling to a 2 k × 2 k matrix, and then multiplication of both dimensions with a squared-sine function before a two-dimensional Fourier transformation and phase correction. Phase correction parameters of the previous experiment were used and were carefully visually controlled and manually adopted for the relevant cross peaks in both dimensions for each measurement. After automatic baseline corrections in both dimensions, integration borders were set uniformly by using the same integration file with the same range, which were set as follows:Aloin (CH-10): 4.71 to 4.45 ppm in F2 (^1^H) and 45.21 to 43.84 ppm in F1 (^13^C);Aloin (CH-6): 7.62 to 7.49 ppm in F2 (^1^H) and 136.90 to 135.69 ppm in F1 (^13^C);Sennosides (CH-10): 5.14 to 4.86 ppm in F2 (^1^H) and 55.02 to 53.62 ppm in F1 (^13^C);Sennoside A (CH-6): 7.81 to 7.61 ppm in F2 (^1^H) and 135.95 to 135.22 ppm in F1 (^13^C);Sennoside B (CH-6): 7.66 to 7.40 ppm in F2 (^1^H) and 135.32 to 134.48 ppm in F1 (^13^C);Sennoside A_1_ (CH-6): 7.43 to 7.31 ppm in F2 (^1^H) and 134.57 to 134.09 ppm in F1 (^13^C).

### 3.7. Method Validation

The method was validated for linearity, repeatability, precision, accuracy, specificity, and limit of quantitation. The evaluation of linearity was achieved by establishing calibration curves over a range of at least 80 to 120% of the measured concentrations. Here, 7-point calibration curves were created and expressed as linear functions. The lowest concentration of the calibration curve was defined as the limit of quantitation. Precision measurements included intra- and inter-day precision, as well as repeatability, and were accomplished in the following way: For intra-day precision, six samples were prepared, and each sample was measured once. Inter-day precision was assessed by the preparation of another six samples in one of the following days. For repeatability, one sample was prepared and measured six-fold. The accuracy of the total sennoside assay was assessed by comparison of the obtained values with the values given on the certificate of analysis of the senna leaflets or with the amount of sennosides indicated for the retard tablets, respectively. The accuracy of the single sennoside assay was accomplished by measuring the total sennoside content of a sample worked up for single sennoside analysis, which thus contained only the dicarboxylic sennosides. The obtained value was subsequently compared to the sum of sennosides A, B, and A_1_ (determined in the precision measurements).

## 4. Conclusions

In this study, we present two quantitative 2D NMR methods for the quality control of senna leaflets, pods, and commercial preparations. In our first method, we quantified the total amount of the value-determining dianthrone glycosides in five minutes of measurement time. This was achieved by band-selective HSQC and by focusing on the 10–10’ bond linking the two monomers. Thus, the most specific (and at the same time stability-indicating) signal for differentiation from monomeric anthranoids was selected. Validation of the method revealed excellent values for accuracy, precision, and linearity and proved qNMR once more as an ideal method for the evaluation of medicinal plants. By applying band-selective experiments, measurement times of UHPLC methods were achieved, thus rendering two-dimensional techniques suited for routine analysis. With the use of aloin as the surrogate standard, moreover, the use of original reference material can be avoided. This is even more important as sennosides are not only cost-intensive reference standards, but also are characterised by their low stability in solution.

In our second method, the major sennosides A, B, and A_1_ were quantified, this time using the CH-6 (and CH-6’) signal of the compounds. Showing different chemical shift values in the ^1^H and ^13^C dimension, all three compounds could be quantified side by side. Though the method also showed good linearity and accuracy, precision measurements revealed high RSD values for sennosides B and A_1_, which were caused by a low resolution and/or concentration, respectively. This could be improved with longer measurement times and an increased number of increments, thereby enhancing the resolution as well as the signal intensity. Alternatively, instruments with higher field strengths can be employed, thus maintaining, or even reducing, the current measurement time of 31 min, which is still acceptable.

Of course, there are other techniques, e.g., LC-MS/MS, which show a much higher grade in sensitivity and are therefore the methods of choice for trace analysis or precise quantitation of minimum amounts. For the quality control of herbal drugs and products measuring in such small concentrations is usually not necessary. Here, other criteria are of greater importance, such as giving a precise and comprehensive value for the content of bioactive components. Not only, but especially, for this purpose, the herein presented determination of the total sennoside content by quantitative NMR is an ideal example. Moreover, other research questions, in which a sum of similar constituents act towards a biological effect, can be addressed with our methodology. One such example is our work on luo han guo, a natural sweetener, where we selectively quantified the sweet-tasting constituents among other structurally related bitter or non-tasting compounds [27]. This was achieved by targeting the crucial structural feature for the sweet taste, in this case, a specific α-hydroxy group. Hence, the determination of other compound (sub)classes via characteristic and activity-relevant structural features, e.g., pharmacophores, are potential applications for our approach.

## Figures and Tables

**Figure 1 molecules-27-07349-f001:**
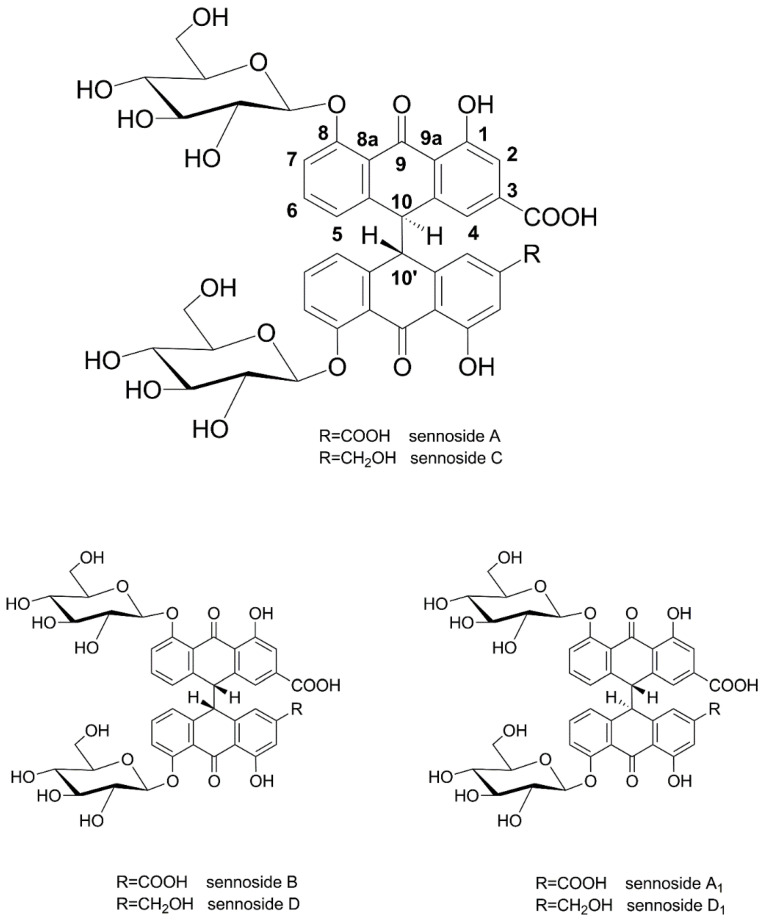
Chemical structures of sennosides A, B, C, D, A_1_, and D_1_.

**Figure 2 molecules-27-07349-f002:**
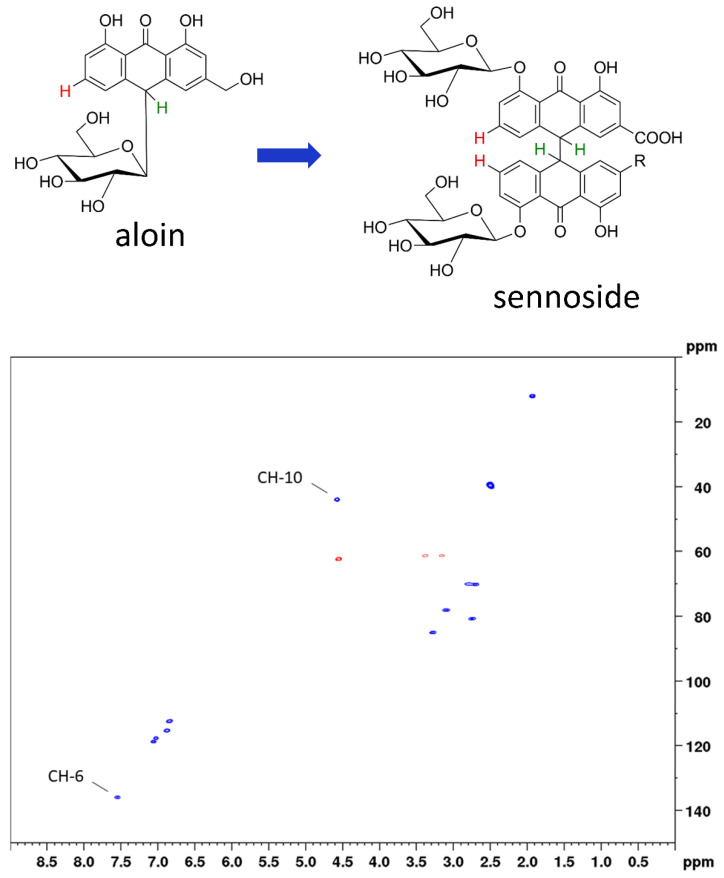
**Top**: chemical structures of surrogate standard aloin and the assayed sennosides with the relevant protons marked in green (H-10 for total sennoside content) and red (H-6 for the content of sennosides A, B, and A_1_). **Bottom**: HSQC diagram of aloin indicating the respective cross peaks used for calibration.

**Figure 3 molecules-27-07349-f003:**
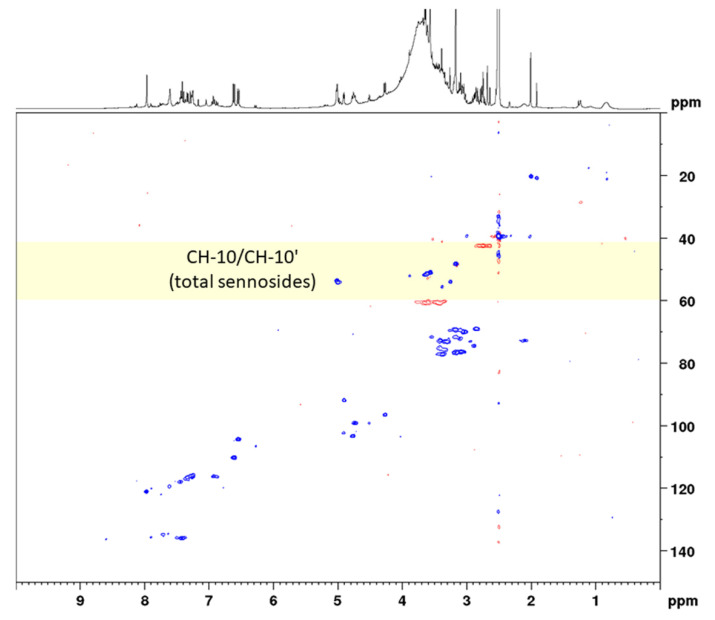
Complete spectral range HSQC diagram of senna solid-phase extract (1 g pods) in DMSO-*d*_6_ with delays optimized for 138 Hz. The ^13^C spectral range used for band-selective measurements is marked in yellow.

**Figure 4 molecules-27-07349-f004:**
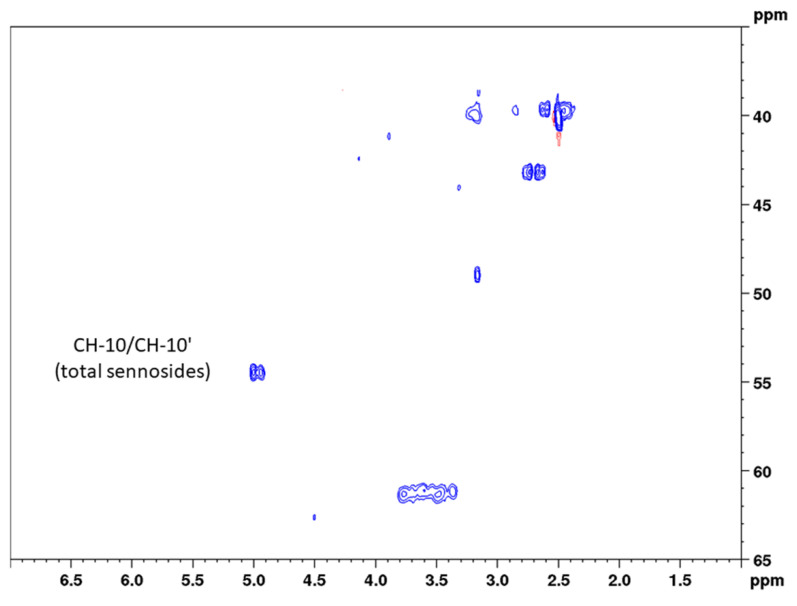
Band-selective HSQC diagram of senna solid-phase extract (1 g pods) in DMSO-*d*_6_ with delays optimized for 138 Hz in the range of 1.0 to 7.0 ppm (^1^H) and 35 to 65 ppm (^13^C).

**Figure 5 molecules-27-07349-f005:**
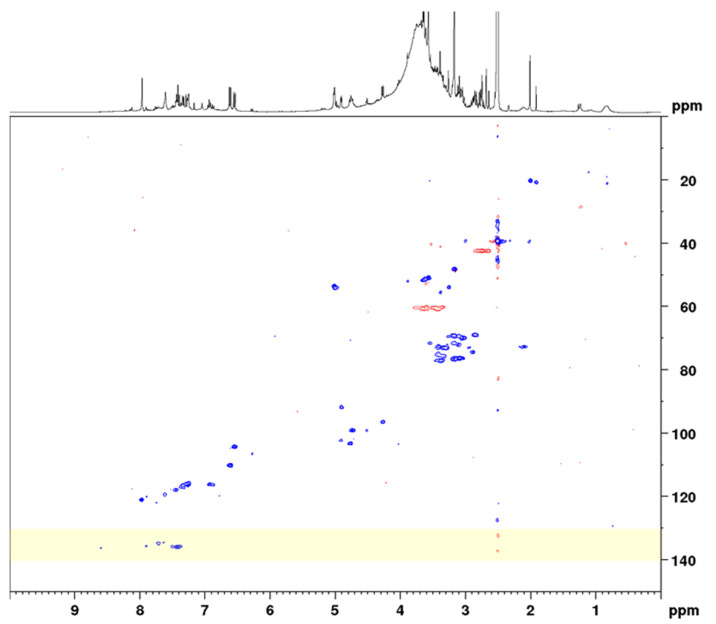
Complete spectral range HSQC diagram of senna solid-phase extract (1 g pods) in DMSO-*d*_6_ with delays optimized for 163 Hz. The ^13^C spectral range used for band-selective measurements is marked in yellow.

**Figure 6 molecules-27-07349-f006:**
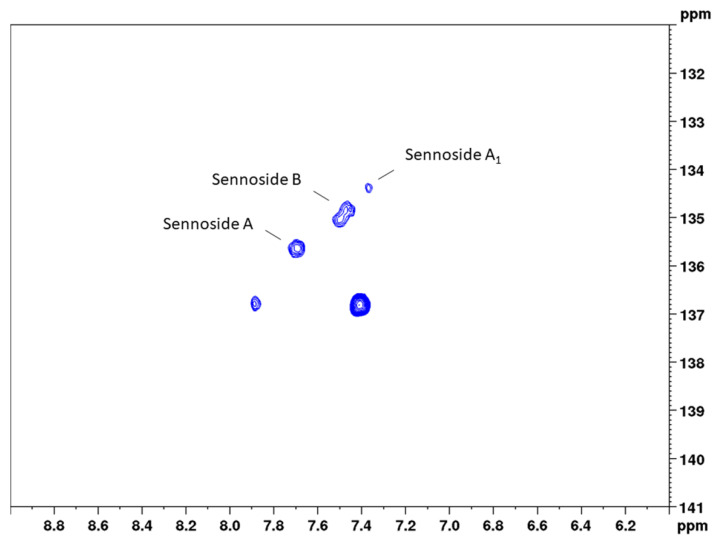
Band-selective HSQC diagram of senna solid-phase extract (1 g pods) in DMSO-*d*_6_ with delays optimized for 163 Hz in the range of 6.0 to 9.0 ppm (^1^H) and 131 to 141 ppm (^13^C).

**Table 1 molecules-27-07349-t001:** Regression equations for the surrogate standard aloin, coefficients of determination (*R*^2^), and limit of quantification (LoQ).

Cross-Correlation Signal	Regression Equation	*R* ^2^	LoQ
CH-10 (total sennosides)	*y* = 6,213,364 *x* − 1,164,109	0.9955	0.500 mmol/L
CH-6 (sennoside A, B, A_1_)	*y* = 69,176,745 *x* − 43,389,229	0.9956	0.875 mmol/L

**Table 2 molecules-27-07349-t002:** Repeatability, intra-day, and inter-day precision of the total sennoside content (calculated as sennoside B) and sennosides A, B, and A_1_. Results are given in % weight of dried plant material; standard deviation is indicated in parenthesis.

Compound Class	Repeatability	Intra-Day 1	Intra-Day 2	Inter-Day
Total sennosides	4.611 (0.103)	4.702 (0.133)	4.812 (0.151)	4.757 (0.147)
Sennoside A	1.073 (0.058)	0.984 (0.069)	0.906 (0.072)	0.945 (0.060)
Sennoside B	1.303 (0.074)	1.095 (0.106)	1.046 (0.121)	1.071 (0.107)
Sennoside A1	0.349 (0.037)	0.206 (0.028)	0.222 (0.016)	0.214 (0.030)

**Table 3 molecules-27-07349-t003:** Repeatability, intra-day, and inter-day precision of the total sennoside content (calculated as sennoside B) and sennosides A, B, and A_1_. Results are given in % weight of dried plant material; standard deviation is indicated in parenthesis.

Product	qNMR	Reference Value	Recovery Rate
Senna leaflets	2.42 ± 0.08%	2.35 ± 0.15%	103.0%
Retard tablets	12.77 ± 0.85 mg	13 mg	98.5%

## Data Availability

Not applicable.

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
