# Peer review of "Determination of Total Sennosides and Sennosides A, B, and A1 in Senna Leaflets, Pods, and Tablets by Two-Dimensional qNMR"

_molecules, 2022, doi:10.3390/molecules27217349_

Round 1

Reviewer 1 Report

This is an interesting work and can be published in this journal.

Author Response

Reviewer 1:

This is an interesting work and can be published in this journal.

A:

Thank you very much for your nice comment!

Reviewer 2 Report

The authors determinatied quantified of sennosides by wo-dimensional qNMR method. They used  band-selective HSQC experiments to obtain the  information obout 10-10' bonds sennosides. This subject is relevant to gain a  isolation and analysise of structure of natural products. I believe the authors made a good scientific work. However, the introduction and discussion do not include citations from relevant articles by key scientists in the field of structure of natural products and two-dimensional heteronuclear correlation NMR method. This fact may be create an erroneous impression about the low awareness of the authors. In this regard, please cite the works from the list of publications doi below:
1. professor S Berger
10.1002/ciuz.201800860
10.1002/ciuz.201970202
10.1002/ciuz.201600762
2. other authors
10.1007/s00775-021-01869-5
10.1016/j.xphs.2021.01.001
10.1007/s11172-017-1701-3
10.1007/s00723-019-01178-w
Minor issues:
- please discuss the possibility of applying registration of non-uniform sampling for your task
-please increase the font size in the legends of the axes in figures 2,3,4

Author Response

Reviewer 2:

The authors determinatied quantified of sennosides by wo-dimensional qNMR method. They used  band-selective HSQC experiments to obtain the  information obout 10-10' bonds sennosides. This subject is relevant to gain a  isolation and analysise of structure of natural products. I believe the authors made a good scientific work. However, the introduction and discussion do not include citations from relevant articles by key scientists in the field of structure of natural products and two-dimensional heteronuclear correlation NMR method. This fact may be create an erroneous impression about the low awareness of the authors. In this regard, please cite the works from the list of publications doi below:
1. professor S Berger
10.1002/ciuz.201800860
10.1002/ciuz.201970202
10.1002/ciuz.201600762
2. other authors
10.1007/s00775-021-01869-5
10.1016/j.xphs.2021.01.001
10.1007/s11172-017-1701-3
10.1007/s00723-019-01178-w

Dear Reviewer,

Thank you very much for honouring our work as well as the suggested publications. We had a look on all seven articles; however, none of the articles deals with quantitative NMR or with the investigated sennosides or other anthranoids or anthranoid drugs used in phytotherapy or phytopharmacology. In example, 10.1002/ciuz.201800860 deals with the monoterpenoid pulegon, which is effective against fleas, 10.1002/ciuz.201970202 is the promotion for a book, and 10.1002/ciuz.201600762 is about artemisinin, a sesquiterpenoid for the treatment of malaria. 

The other four suggested articles are even further away from the topic of our study, dealing with cyanocobalamin chlorination (10.1007/s00775-021-01869-5), conformations of carbamazepine (10.1016/j.xphs.2021.01.001), tautomerism studies (10.1007/s11172-017-1701-3), or ninhydrin derivatives (10.1007/s00723-019-01178-w).

Therefore, we see neither the purpose nor the justification to include these references into our manuscript, though they for sure are relevant to the respective research fields.

Minor issues:
- please discuss the possibility of applying registration of non-uniform sampling for your task

We included a statement a reference [29] on non-uniform sampling into our discussion (line 169 to 173 in the marked version).

-please increase the font size in the legends of the axes in figures 2,3,4

Figure 2,3, and 4 were changed so that axes and details are better readable. Thereby, figures 3 and 4 were each splitted into two figures, now named figures 3 to 6 instead of figure 3 (left and right) and figure 4 (left and right), respectively.

Reviewer 3 Report

 In this work, Serhat Sezai Çiçek et al. established 2D-qNMR methods to quantify total sennosides and perform single analysis for sennosides A, B, and A1. The specificity is higher than current techniques, though no novel technology was developed. Overall, the data is convincing and logic is clear in this paper. However, there are some minor concerns and comments, requesting some revisions/updates discussed below and the manuscript should be published after that.

1.      Number the carbons for the structures of sennosides in Figure 1.

2.      Please compare the 2D-qNMR sensitivity with widely used LC-MS/MS, which gives the readers a general sense of this technique and promotes the dissemination.

3.      The authors should give a more comprehensive comments of LC-MS/MS vs. 2D-qNMR.

4.      Currently, as an emerging technique, the LC-NMR technique might give the authors option to solve the low resolution of HSQC signals between sennosides B and A1.

5.      Please provide more discussions regarding the potential applications of the technique.

Author Response

Reviewer 3:

In this work, Serhat Sezai Çiçek et al. established 2D-qNMR methods to quantify total sennosides and perform single analysis for sennosides A, B, and A1. The specificity is higher than current techniques, though no novel technology was developed. Overall, the data is convincing and logic is clear in this paper. However, there are some minor concerns and comments, requesting some revisions/updates discussed below and the manuscript should be published after that.

Dear Reviewer,

Thank you very much for honouring our work. Please find our answers below:

  1. Number the carbons for the structures of sennosides in Figure 1.

Carbon numbering was inserted into Figure 1.

  1. Please compare the 2D-qNMR sensitivity with widely used LC-MS/MS, which gives the readers a general sense of this technique and promotes the dissemination.

We included a paragraph into the conclusion section where we mention the difference to LC-MS/MS (line 520 to 526 in the marked version).

  1. The authors should give a more comprehensive comments of LC-MS/MS vs. 2D-qNMR.

See point 2.

  1. Currently, as an emerging technique, the LC-NMR technique might give the authors option to solve the low resolution of HSQC signals between sennosides B and A1.

LC NMR coupling is of course also a technique, which would deliver the information given by the presented HSQC methods. However, due to coupling with chromatographic separation, sample amounts are much lower as presented in the manuscript and therefore the technical equipment requires high magnetic field magnets and is very expensive. Thus, no serious comparison can be done.

  1. Please provide more discussions regarding the potential applications of the technique.

We provided a few sentences on potential applications at the end of our manuscript (line 526 to 534 in the marked version).